# RETHINKING SELF-SUPERVISED LEARNING: AN INSTANCE-WISE SIMILARITY PERSPECTIVE

## ABSTRACT

This paper studies self-supervised learning from the perspective of instance-wise similarity (IwS), characterized by the pairwise similarity matrix among all instances. Ideally, the IwS matrix in the representation space should closely mirror that in the input space so that the learned representations retain their discriminative power and account for semantic similarities. This perspective not only allows us to understand diverse existing self-supervised learning methodologies better but also uncovers a notable limitation within current approaches: the discrepancy between IwS matrices in the input and representation spaces. Indeed, many established methods, including SimCLR and MoCo v3, implicitly assume that the IwS matrix within the representation space is an identity matrix, even when the IwS matrix in the input space may deviate from this form. Inspired by this observation, we introduce sparse contrastive learning, a new approach that learns an appropriately sparse IwS matrix within the representation space instead of presuming an identity IwS matrix. Our comprehensive experiments conducted on ImageNet and CIFAR datasets substantiate the superior performance of our method in comparison to other state-of-the-art methods.

## 1 INTRODUCTION

Self-supervised learning excels in acquiring transferable representations that remain invariant to various augmentations, and it has gained significant popularity within the machine learning community (Chen et al., 2020a; He et al., 2020; Caron et al., 2020; Grill et al., 2020; Zbontar et al., 2021). This approach has found extensive application in diverse tasks, including multimodality learning, object detection, and segmentation (Radford et al., 2021; Li et al., 2022; Xie et al., 2021; Wang et al., 2021; Yang et al., 2021; Zhao et al., 2021). The pursuit of a deeper understanding of self-supervised learning stands as an important research avenue (Arora et al., 2019; Wang & Isola, 2020; Wang & Liu, 2021). In this paper, we introduce a novel perspective that rethinks self-supervised learning.

Specifically, we approach self-supervised learning from the perspective of Instance-wise Similarity (IwS), characterized by the pairwise similarity matrix among all instances. Figure 1 provides a visual representation of the IwS matrix, which is a binary matrix indicating the semantic similarity between any two arbitrary instances. This matrix can be computed in both the input and representation (embedding) spaces. Ideally, the IwS matrix in the representation space should closely mirror that in the input space, ensuring that the learned representations maintain their discriminative abilities and effectively capture semantic similarities. However, obtaining IwS in the input space is generally unfeasible due to the absence of labels/supervision, making it impossible to use IwS directly for representation learning.

Contrastive learning methods, including SimCLR (Chen et al., 2020a), MoCo v3 (Chen et al., 2021), and various other variants (Garrido et al., 2023; Ge et al., 2023; Chuang et al., 2020; Dwibedi et al., 2021; Robinson et al., 2021), address this challenge by implicitly assuming the IwS matrix to be an identity matrix. These methods aim to align positive pairs, comprising augmented image pairs generated from the same image, while simultaneously repelling negative pairs, which can consist of any two distinct images from the training dataset. In essence, only the diagonal entries of the IwS matrix in the representation space are set to 1, as depicted in Figure 1(c). However, this assumption may deviate from the underlying IwS matrix in the input space. For instance, both the (1,3) and (3,1) entries are 1 instead of 0 because the 1st and 3rd images are semantically similar (both depicting

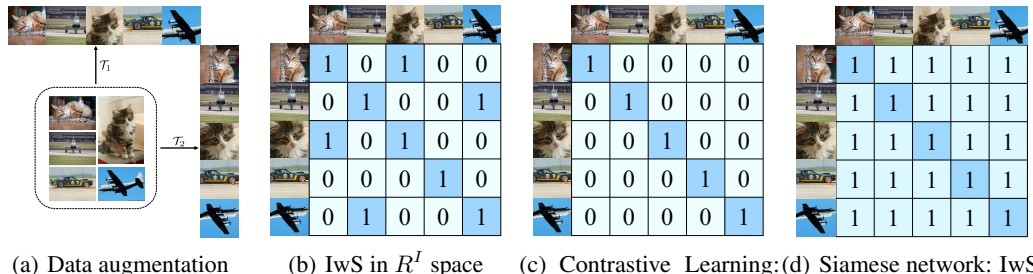

(a) Data augmentation  (b) IwS in $R^I$ space  (c) Contrastive Learning: IwS in $R^S$ space  (d) Siamese network: IwS in $R^S$ space

Figure 1: Visualizing the IwS matrix in the input space ($R^I$), and the representation space ($R^S$) for the Siamese network and contrastive learning. The 1/0 indicates the pairwise semantic similarity/difference. (a) illustrates data augmentation. (b) visualizes the IwS matrix in the input space. Some off-diagonal entries are 1. For example, both the (3, 1) and (1, 3) entries are 1, indicating that the 1st and 3rd images share the same semantic/label (cat). (c) shows the IwS matrix for contrastive learning methods such as SimCLR (Chen et al., 2020a), which is implicitly assumed as an identity matrix. (d) shows the IwS matrix for the Siamese network in the representation space, which is an all-one matrix.

cats), as illustrated in Figure 1(b). Therefore, simply assuming an identity structure can potentially hinder the algorithm from learning semantically meaningful representations, resulting in reduced performance in downstream tasks.

In contrast, a Siamese network (Hadsell et al., 2006) focuses solely on aligning positive pairs, which can potentially lead to a constant solution where all representations collapse into one single point (Chen & He, 2021). This issue is commonly referred to as the "collapse problem" (Jing et al., 2022). When this occurs, the IwS matrix in the representation space becomes an all-one matrix, as depicted in Figure1(d). Certain variants, like BYOL (Grill et al., 2020) and SimSiam (Chen & He, 2021), introduce an asymmetric network architecture to mitigate this problem. However, lacking explicit constraints on off-diagonal entries, these methods fail to fully exploit the sparse structure inherent in the IwS matrix.

Therefore, the Siamese network and contrastive learning represent two opposing extremes: Contrastive learning assumes the IwS matrix to be an identity matrix (or extremely sparse), while the Siamese network does not leverage any sparse structure within the IwS matrix, as illustrated in Figures 1(c) and 1(d). To strike a balance between these two extremes, we introduce a sparsity penalty/loss that encourages an appropriate level of sparsity in the IwS matrix within the representation space. Leveraging this sparsity penalty, we present Sparse Contrastive Learning (Sparse CL), a novel self-supervised learning approach, for representation learning that fully utilizes the sparse structure inherent in the IwS matrix. Our extensive experimental results on ImageNet-100/1k and CIFAR-10/100 datasets empirically demonstrate the effectiveness of our proposed method.

Our primary contributions are as follows. (i) We study self-supervised learning from the perspective of IwS, providing a novel framework. (ii) We pinpoint critical issues in both the Siamese network, associated with an overly dense IwS matrix, and existing contrastive learning methods, which assume identity IwS matrices. (iii) We introduce Sparse CL, a novel self-supervised learning approach, to learn an appropriately sparse IwS matrix in the representation space, resulting in improved representations. (iv) Our proposed Sparse CL achieves state-of-the-art performance in linear evaluation on several benchmark datasets.

## RELATED WORK

**Dimension Contrastive Learning and Others**  Dimension-contrastive methods penalize the off-diagonal terms of the covariance matrix of the embeddings rather than the IwS matrix (Garrido et al., 2023). For instance, Barlow Twins (Zbontar et al., 2021) and VICReg (Bardes et al., 2022) make the covariance matrix as close to an identity one as possible, while W-MSE(Ermolov et al., 2021) and Zero-CL (Zhang et al., 2022) directly whiten the covariance matrix. Clustering-based self-supervised methods (Caron et al., 2018; 2020; 2021) are based on the prototype-wise or cluster-wise similarity.

**Weak Supervision** Inspired by the success of supervised contrastive learning (Khosla et al., 2020), weak supervision has been adopted in self-supervised learning. For instance, AdpCLR (Zhang et al., 2021) initiates by identifying the top-$k$ nearest samples in the representation space and subsequently employs this information to construct an adjacency matrix, which serves as a form of supervision. In pursuit of improved supervision, WCL (Zheng et al., 2021a; Chen et al., 2022a) takes a step further by enforcing symmetry in the nearest neighbor graph. Other approaches by (Huynh et al., 2022) and (Chen et al., 2022b) utilize detected false negative samples as a means of supervision in self-supervised learning. However, these methods grapple with the limitation of weak or even inaccurate supervision, resulting in suboptimal performance.

## 2 METHODOLOGY

We first review the fundamentals of self-supervised learning in Section 2.1. Subsequently, in Section 2.2, we introduce the IwS perspective to reevaluate self-supervised learning. Section 2.3 presents Sparse CL, a novel self-supervised approach designed to learn an appropriately sparse IwS matrix in the representation space and thus better representations.

### 2.1 PRELIMINARY: SELF-SUPERVISED LEARNING

To learn invariant representations across various augmentations (Chen et al., 2020a; He et al., 2020), most self-supervised learning methods maximize the similarity of positive pairs using Siamese networks (Hadsell et al., 2006; Grill et al., 2020; Chen & He, 2021). A positive pair consists of two augmentations $\mathcal{T}_1$ and $\mathcal{T}_2$ applied to the same sample. Specifically, given a set of samples in the input space, $\mathbf{X} = \{x_1, x_2, ..., x_n\}$, each sample in $\mathbf{X}$ is augmented to create two views, denoted as $x_i^a = \mathcal{T}_1(x_i)$ and $x_i^b = \mathcal{T}_2(x_i)$. Most self-supervised learning methods (Grill et al., 2020; Dwibedi et al., 2021; He et al., 2020; Chen & He, 2021; Chen et al., 2021) swap the positive pair $(x_i^a, x_i^b)$ to obtain a second pair $(x_i^b, x_i^a)$, and then feed both pairs to the online encoder ($f_\theta(\cdot)$) and target encoder($f_\xi(\cdot)$) to obtain the feature embeddings: $q_i^a = f_\theta(x_i^a)$, $q_i^b = f_\theta(x_i^b)$, $k_i^a = f_\xi(x_i^a)$, $k_i^b = f_\xi(x_i^b)$.

To obtain invariant representations, $(q_i^a, k_i^b)$ and $(q_i^b, k_i^a)$, of the positive pairs, mean squared error (MSE) is a commonly used loss function to align their $\ell_2$ normalized representations on the hypersphere (Grill et al., 2020; Ermolov et al., 2021). Specifically, the alignment loss is defined as:

$$\mathcal{L}_A = \frac{1}{2} \sum_{i=1}^n \left( \left\| \frac{q_i^a}{\|q_i^a\|} - \frac{k_i^b}{\|k_i^b\|} \right\|_2^2 + \left\| \frac{q_i^b}{\|q_i^b\|} - \frac{k_i^a}{\|k_i^a\|} \right\|_2^2 \right), \tag{1}$$

In this paper, we revisit self-supervised learning from the perspective of instance-wise similarity, with its formal definition introduced in the next section.

### 2.2 INSTANCE-WISE SIMILARITY

We define instance-wise similarity as a function that maps two arbitrary instances in any Euclidean space, including both the input and representation spaces, to a semantic similarity indicator. A value of 1 indicates the same semantic meaning, while a value of 0 indicates different semantics. Specifically, let

$$w_{ij} : \mathbb{R}_i^m \times \mathbb{R}_j^m \to 0/1,$$

be the semantic similarity indicator between the $i$-th and $j$-th instances in $\mathbb{R}^m$.

**Input Space** Given a positive set pair

$$\left( \mathbf{X}^a = \{x_1^a, x_2^a, \ldots, x_n^a\}, \mathbf{X}^b = \{x_1^b, x_2^b, \ldots, x_n^b\} \right),$$

where each sample pair $(x_i^a, x_i^b)$ forms a positive pair in the input space, the instance-wise similarity (IwS) can be characterized by a binary matrix:

$$\mathbf{W}^{ab} = \begin{pmatrix} w_{11} & w_{12} & \cdots & w_{1n} \\ w_{21} & w_{22} & \cdots & w_{2n} \\ \cdots & \cdots & \cdots & \cdots \\ w_{n1} & w_{n2} & \cdots & w_{nn} \end{pmatrix}, \tag{2}$$

where $w_{ij}$ indicates the semantic similarity between $x_i^a$ and $x_j^b$. As previously defined, it takes a value of either 1 when $c(x_i^a) = c(x_j^b)$ or 0 when $c(x_i^a) \neq c(x_j^b)$, where $c(x)$ is a function indicating the underlying class of sample $x$ (Arora et al., 2019). In self-supervised learning, the semantic information is invariant to different augmentations, so $x_i^a$ and $x_i^b$ share the same semantics, i.e., $w_{ii} = 1$. For the off-diagonal entries $w_{ij}, i \neq j$, they can be either 1 or 0. Thus, the IwS matrix exhibits sparsity, and the level of sparsity depends on the number of classes and the data distribution. Generally, as the number of classes increases or the data distribution becomes more balanced[1], the sparsity of the IwS matrix becomes more pronounced.

**Representation Space**   Given the positive set pair

$$\left( \mathbf{Q}^a = \{q_1^a, q_2^a, ..., q_n^a\}, \ \mathbf{K}^b = \{k_1^b, k_2^b, ..., k_n^b\} \right)$$

or $\mathbf{Q}^b$ and $\mathbf{K}^a$, the IwS matrix is a binary matrix $\hat{\mathbf{W}}^{ab} = (\hat{w}_{ij})^{n \times n}$, where $\hat{w}_{ij}$ signifies the semantic similarity between $q_i^a$ and $k_j^b$. It takes on a value of 1 when $c(q_i^a) = c(k_j^b)$ (indicating they belong to the same class) and 0 when $c(q_i^a) \neq c(k_j^b)$ (indicating different classes). Ideally, the IwS matrix in the representation space should closely resemble the one in the input space to ensure that the learned representations maintain their discriminative power and capture semantic similarities, i.e., $\mathbf{W}^{ab} = \hat{\mathbf{W}}^{ab} = \hat{\mathbf{W}}^{ba}$. In self-supervised learning, the IwS matrix in the input space remains unknown due to the absence of labels or supervision, making it challenging to pick an appropriate IwS matrix in the representation space. To address this challenge, most self-supervised approaches adopt simple, albeit less accurate, strategies.

**The Issue with Siamese Networks**   Siamese networks (Hadsell et al., 2006) employs a symmetric network to align positive pairs. However, there are trivial collapsing solutions (Chen & He, 2021), corresponding to an all-one IwS matrix in the representation space, i.e., $\mathbf{W}^{ab} = \hat{\mathbf{W}}^{ab} = \mathbf{1}_{n \times n}$. Some variants, such as BYOL (Grill et al., 2020) and SimSiam (Chen & He, 2021), propose additional techniques like employing an extra predictor, momentum updates, and stop-gradient operators to mitigate this collapsing issue. However, the lack of explicit constraints on the off-diagonal positions makes it insufficient to fully exploit the sparse structure of the underlying IwS matrix.

**The Issue with Contrastive Learning**   Contrastive learning methods, such as SimCLR (Chen et al., 2020a) or MoCo v3 (Chen et al., 2021), operate by aligning positive pairs and repelling negative pairs. These methods implicitly assume that the IwS matrix in the representation space is an identity matrix, i.e., $\hat{\mathbf{W}}^{ab} = \hat{\mathbf{W}}^{ba} = \mathbf{I}_n$. However, as discussed earlier, the IwS matrix in the input space may deviate from the identity matrix. Consequently, using the identity IwS matrix can compromise the discriminative power of learned representations, ultimately resulting in suboptimal performance in downstream tasks. In essence, contrastive learning adopts an overly aggressive approach by using an extremely sparse IwS matrix in the representation space.

### 2.3   Sparse Contrastive Learning

In this section, we introduce a novel self-supervised approach aimed at learning an appropriately sparse IwS matrix in the representation space. To achieve this, we formulate a sparsity loss/penalty that can encourage the desired sparsity level in the IwS matrix for representation learning. While one direct idea might be to apply the $\ell_1$ norm to $\hat{\mathbf{W}}^{ab}$ and $\hat{\mathbf{W}}^{ba}$ to construct the loss objective, the non-differentiability of $\hat{w}_{ij}$ in the IwS matrix poses a challenge for end-to-end training.

Instead, we propose to use the cosine similarity between $q_i$ and $k_j$ as $\hat{w}_{ij}$:

$$\hat{w}_{ij} = \sigma \left( \frac{q_i \cdot k_j - t_0}{\tau} \right) = \frac{1}{1 + \exp \left( \frac{-(q_i \cdot k_j - t_0)}{\tau} \right)}, \tag{3}$$

---

[1] Assuming there are $K$ distinct classes, and the proportions for each class are denoted as $p_1, p_2, \ldots, p_K$, we have $p(w_{ij} = 1, i \neq j) = \sum_{i=1}^{K} p_i^2 \geq \left( \sum_{i=1}^{K} p_i \right)^2 / K = \sum_{i=1}^{K} (1/K)^2$. Consequently, with a more balanced data distribution, the probability of $p(w_{ij} = 1, i \neq j)$ diminishes, resulting in increased sparsity within the IwS matrix.

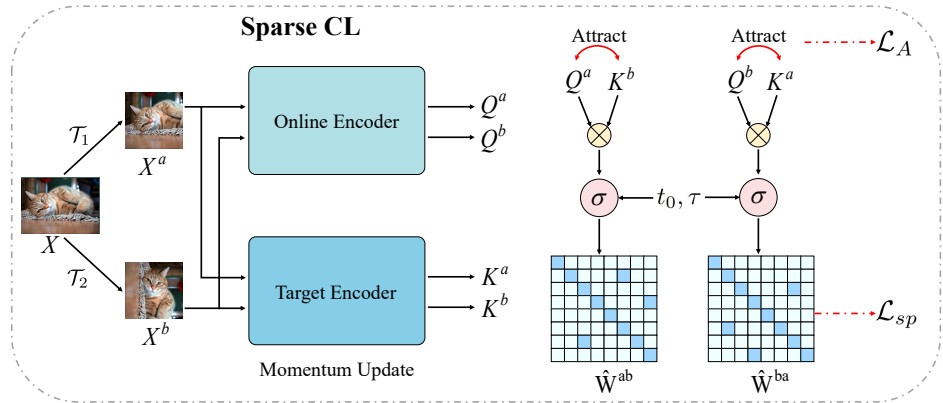

Figure 2: The overall illustration of our proposed method Sparse CL, which consists of an online encoder, and a target encoder. $\otimes$ and $\sigma$ represent dot product and sigmoid activation, respectively. During the pre-training phase, we extract query and key embeddings from the online encoder and target encoder to construct two element-wise positive pairs $(\mathbf{Q}^a, \mathbf{K}^b)$ and $(\mathbf{Q}^b, \mathbf{K}^a)$, base on which the IwS in the representation space, e.g., $\hat{\mathbf{W}}^{ab}$ and $\hat{\mathbf{W}}^{ba}$, are estimated in a differentiable manner. Then we achieve sparsity in the representation space by imposing $\ell_1$ norm on the estimated IwS.

where $\cdot$ denotes the inner product, $t_0$ is a thresholding constant, and $\tau$ is the temperature parameter. We choose a relatively small temperature $\tau = 0.1$ to make $\hat{w}_{ij}$ close to 1 or 0. Specifically, $\hat{w}_{ij} \approx 1$ when $q_i \cdot k_j > t_0$, and $\hat{w}_{ij} \approx 0$ when $q_i \cdot k_j < t_0$. Finally, the proposed sparsity loss or penalty is:

$$\mathcal{L}_{\text{sp}} = \frac{1}{2} \left( \sum_{\substack{\hat{w}_{ij} \in \hat{\mathbf{W}}^{ab} \\ i \neq j}} \|\hat{w}_{ij}\|_1 + \sum_{\substack{\hat{w}_{ij} \in \hat{\mathbf{W}}^{ba} \\ i \neq j}} \|\hat{w}_{ij}\|_1 \right) \tag{4}$$

When combined with the alignment loss described in Equation 1, the overall objective function for self-supervised learning becomes:

$$\min_{\theta} \mathcal{L}_A + \lambda \, \mathcal{L}_{\text{sp}}, \tag{5}$$

where $\lambda$ serves as a regularization parameter that controls the trade-off between the alignment loss and the sparsity loss. We refer to this method as Sparse Contrastive Learning, aka Sparse CL. Figure 2 provides a visual summary of the proposed approach.

## 3 EXPERIMENTS

This section empirically validates the performance of Sparse CL. More specifically, Section 3.1 presents the results of experiments conducted on four well-established benchmarks frequently used in self-supervised learning: CIFAR-10, CIFAR-100, ImageNet-100, and ImageNet-1k (Zhang et al., 2022). To provide further insights into the proposed approach, additional analyses and visualizations are carried out in Section 3.2. We will release the PyTorch code publicly.

**Baseline Methods**  We benchmark Sparse CL against a set of baseline methods, including Zero-CL (Zhang et al., 2022), DeepCluster V2 (Caron et al., 2020), DINO (Caron et al., 2021), MoCo V2 (Chen et al., 2020b), NNCLR (Dwibedi et al., 2021), ReSSL (Zheng et al., 2021b), SimCLR (Chen et al., 2020a), SimSiam (Chen & He, 2021), SwAV (Caron et al., 2020), VICReg (Bardes et al., 2022), W-MSE (Ermolov et al., 2021), Barlow Twins (Zbontar et al., 2021), BYOL (Grill et al., 2020), and MoCo v3 (Chen et al., 2021). Notably, for CIFAR-10/100 and ImageNet-100 datasets, we cite results from solo-learn (da Costa et al., 2022), which reports superior performance compared to the original papers or other third-party sources.

We further explore the proposed Sparse CL from three different perspectives. Firstly, we perform an ablation study by removing the sparsity loss term from Sparse CL, denoted as Sparse CL-s. Next, we evaluate Sparse CL in a supervised setting, where the IwS matrix in the representation space equals that in the input space. Lastly, we examine Sparse CL in the context of transfer learning. For detailed experimental settings, please refer to Appendix A.

Table 1: Linear evaluation performance on CIFAR-10, CIFAR-100 and ImageNet-100.[†]: results by (da Costa et al., 2022), [§]: results by (Zhang et al., 2022). ↑ indicates gains. Best self-supervised baselines are underlined.

| Methods | CIFAR-10 | | CIFAR-100 | | ImageNet-100 | |
|---|---|---|---|---|---|---|
| | Acc@1↑ | Acc@5↑ | Acc@1↑ | Acc@5↑ | Acc@1↑ | Acc@5↑ |
| Zero-CL[§] | 90.81 | 99.77 | 70.33 | 92.05 | 79.26 | 94.98 |
| DeepCluster V2[†] | 88.85 | 99.58 | 63.61 | 88.09 | 75.36 | 93.22 |
| DINO[†] | 89.52 | 99.71 | 66.76 | 90.34 | 74.84 | 92.92 |
| MoCo V2[†] | 92.94 | 99.79 | 69.89 | 91.65 | 78.20 | 95.50 |
| NNCLR[†] | 91.88 | 99.78 | 69.62 | 91.52 | 79.80 | 95.28 |
| ReSSL[†] | 90.63 | 99.62 | 65.92 | 89.73 | 76.92 | 94.20 |
| SimCLR[†] | 90.74 | 99.75 | 65.78 | 89.04 | 77.04 | 94.02 |
| SimSiam[†] | 90.51 | 99.72 | 66.04 | 89.62 | 74.54 | 93.16 |
| SwAV[†] | 89.17 | 99.68 | 64.88 | 88.78 | 74.04 | 92.70 |
| VICReg[†] | 92.07 | 99.74 | 68.54 | 90.83 | 79.22 | 95.06 |
| W-MSE[†] | 88.67 | 99.68 | 61.33 | 87.26 | 67.60 | 90.94 |
| Barlow Twins[†] | 92.10 | 99.73 | 70.90 | 91.91 | 80.38 | 95.28 |
| BYOL[†] | 92.58 | 99.79 | 70.46 | 91.96 | 80.16 | 94.80 |
| MoCo v3[†] | 93.10 | 99.80 | 68.83 | 90.57 | 80.36 | 95.18 |
| Sparse CL-s | 93.12 | 99.86 | 71.45 | 92.92 | 80.06 | 95.66 |
| Sparse CL | **93.45** ↑$_{0.33}$ **99.89** | | **73.09** ↑$_{1.64}$ **93.72** | | **80.98** ↑$_{0.92}$ **95.72** | |

## 3.1 EXPERIMENTAL RESULTS

**Main Results** We conduct experiments on CIFAR-10/100 and ImageNet-100/1k datasets and evaluate all experiments following a linear evaluation protocol. As presented in Tables 1 and 2, our proposed Sparse CL consistently outperforms all prior methods in terms of top-1 accuracy (Acc@1) and top-5 accuracy (Acc@5) across all datasets. Notably, our Sparse CL, when trained for 100 and 200 epochs on the ImageNet-1k dataset, surpasses the best-performing model, NNCLR (Dwibedi et al., 2021), by **1.7** and **2.0** in top-1 accuracy, respectively. Intriguingly, even with just 100 epochs of training, Sparse CL outperforms most self-supervised models trained for 200 epochs. It is worth noting that BYOL (Grill et al., 2020) and MoCo v3 (Chen et al., 2021) share the same network architecture as Sparse CL, with the main distinction being the different constraints applied to the IwS matrix in the representation space.

Table 2: Top-1 Accuracy under linear evaluation on the ImageNet-1k dataset. [†]: results by (Dwibedi et al., 2021), [‡]: results by (Chen et al., 2021), [§]: results by (Zhang et al., 2022). Best self-supervised baseline models are underlined.

| Methods | 100 eps | 200 eps |
|---|---|---|
| SimCLR[†] | 66.5 | 68.3 |
| MoCo v2[†] | 67.4 | 69.9 |
| BYOL[†] | 66.5 | 70.6 |
| SwAV[†] | 66.5 | 69.1 |
| SimSiam[†] | 68.1 | 70.0 |
| NNCLR[†] | 69.4 | 70.7 |
| MoCo v3[‡] | 68.9 | - |
| Barlow Twins[§] | 67.7 | - |
| Zero-CL[§] | 68.9 | - |
| Sparse CL | **71.1** | **72.7** |

Specifically, BYOL does not leverage the sparse structure of the IwS matrix, MoCo v3 implicitly assumes the IwS matrix to be an identity matrix, while Sparse CL utilizes the proposed sparsity penalty. Sparse CL achieves the best performance, providing empirical evidence of the effectiveness of the proposed sparsity penalty.

**Without the Sparsity Loss** Table 1 demonstrates the importance of the sparsity penalty in self-supervised learning. Sparse CL-s, which is Sparse CL without the sparsity penalty, achieves less favorable results in terms of top-1 accuracy (Acc@1) and top-5 accuracy (Acc@5) across CIFAR-10/100 and ImageNet-100 datasets. This suggests that the sparsity penalty plays a crucial role in enhancing performance. Sparse CL outperforms Sparse CL-s by significant margins, achieving improvements of **0.35%**, **2.29%**, and **1.15%** in top-1 accuracy (Acc@1) on CIFAR-10, CIFAR-100, and ImageNet-100 datasets, respectively.

**Supervised Sparse Contrastive Learning** We conduct an experiment where we directly use the IwS matrix in the input space as supervision for the IwS matrix in the representation space. Table 3 collects the results and

Table 3: Linear evaluation performance on CIFAR-10/100 and ImageNet-100. Unsup and Sup are abbreviations for unsupervised and supervised.

| Methods | CIFAR-10 | | CIFAR-100 | | ImageNet-100 | |
|---|---|---|---|---|---|---|
| | Acc@1↑ | Acc@5↑ | Acc@1↑ | Acc@5↑ | Acc@1↑ | Acc@5↑ |
| Sparse CL (Unsup) | 93.45 | **99.89** | 73.09 | **93.72** | 80.98 | 95.72 |
| Sparse CL (Sup) | **94.45** ↑$_{1.00}$ 99.88 | | **74.77** ↑$_{1.68}$ 93.68 | | **81.64** ↑$_{0.66}$ **96.02** | |
| ResNet-18 (Sup) | 92.24 | 99.83 | 68.88 | 90.76 | - | - |

reveals that Sparse CL in the supervised setting outperforms its unsupervised counterpart by **1.1%**,

**2.30%**, and **0.81%** in terms of top-1 accuracy (Acc@1) on CIFAR-10, CIFAR-100, and ImageNet-100 datasets, respectively. Thus when the IwS matrix in the representation space closely mirrors that in the input space, the learned representations indeed retain their discriminative capabilities and capture semantic similarities better.

**Transfer Learning** We fine-tune the entire Sparse CL network on CIFAR-10/100, Oxford-IIIT Pets (Parkhi et al., 2012), and Oxford 102 Flowers (Nilsback & Zisserman, 2008) datasets using the weights pre-trained on ImageNet-1k as an initialization. Table 4 presents the results of transfer learning across these four datasets. Except for the Pets dataset, Sparse CL outperforms other methods, indicating its capability to learn representations that are more transferable.

Table 4: Transfer learning performance using ResNet-50 pretrained with ImageNet-1k. For CIFAR-10/100 datasets, we report Top-1 Accuracy. For Pets and Flowers datasets, we report mean per-class accuracy. [†]: results by (Dwibedi et al., 2021).

| Methods | CIFAR-10 | CIFAR-100 | Pets | Flowers |
|---|---|---|---|---|
| SimCLR[†] | 90.5 | 74.4 | 84.6 | 92.6 |
| BYOL[†] | 91.3 | 78.4 | 90.4 | 96.1 |
| NNCLR[†] | 93.7 | 79.0 | **91.8** | 95.1 |
| Sparse CL | **96.6** | **83.5** | 89.8 | **97.3** |

## 3.2 EXPERIMENTAL ANALYSES

To gain a deeper understanding of Sparse CL, we explore various approaches for implementing the sparsity penalty. Following that, we carry out an array of qualitative and quantitative analyses to elucidate how the proposed sparsity loss enhances representation learning. Additionally, we perform sensitivity analyses on several hyperparameters of the model. These sensitivity analyses are conducted on the CIFAR-100 dataset, with three distinct hyperparameters considered. Training dynamics on the CIFAR-100 dataset and an analysis of the role of asymmetric architecture components in Sparse CL can be found in Appendix B and Appendix C, respectively. We also explore replacing the $\ell_1$ norm with the $\ell_2$ norm in the sparsity loss, and collect the details in Appendix D. Lastly, Appendix E contains a sensitivity analysis conducted with varying batch sizes on the CIFAR-100 dataset.

**On the Sparsity Loss** We explore alternative approaches for constructing the sparsity loss, including applying the Softmax activation, Sparsemax (Martins & Astudillo, 2016), and Gumbel-Softmax (Jang et al., 201t) to the off-diagonal entries of the cosine similarity matrix, followed by using the entropy as our sparsity loss. Table 5 demonstrates that our proposed choice outperforms the other options on CIFAR-10/100 datasets.

Table 5: Linear evaluation performance on CIFAR-10, CIFAR-100 datasets.

| Choices | CIFAR-10 | | CIFAR-100 | |
|---|---|---|---|---|
| | Acc@1↑ | Acc@5↑ | Acc@1↑ | Acc@5↑ |
| Softmax + entropy | 92.76 | **99.90** | 71.72 | 92.98 |
| Sparsemax + entropy | 92.92 | 99.85 | 72.21 | 93.32 |
| Gumbel-Softmax + entropy | 93.03 | 99.88 | 71.75 | 92.93 |
| Sigmoid + $\ell_1$ norm (Our) | **93.45** | 99.89 | **73.09** | **93.72** |

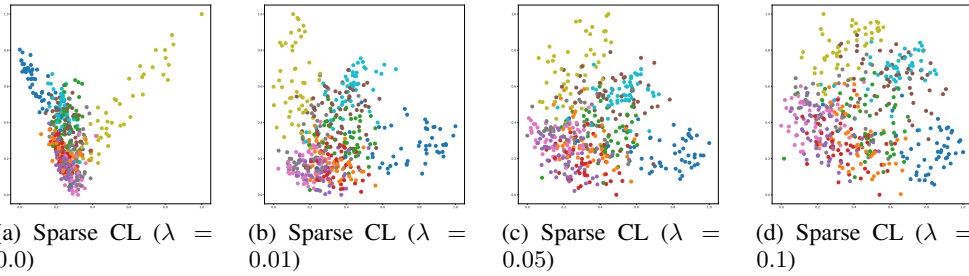

(a) Sparse CL ($\lambda = 0.0$)
(b) Sparse CL ($\lambda = 0.01$)
(c) Sparse CL ($\lambda = 0.05$)
(d) Sparse CL ($\lambda = 0.1$)

Figure 3: PCA visualization on the CIFAR-100 dataset. Classes (0-9) are distinguished by colors. Sparse CL ($\lambda = 0.0$) learns concentrated representations while Sparse CL ($\lambda = 0.1$) is able to learn relatively divergent representations.

**Representation Visualizations** To provide an intuitive understanding of how the proposed sparsity loss affects the distribution of learned representations, we conduct experiments on the CIFAR-100 dataset. We extract the top layer of a ResNet (He et al., 2016) that is trained using Sparse CL and then use these representations for analysis. We project the extracted representations from $\mathbb{R}^{512}$ to $\mathbb{R}^2$ using principal component analysis (PCA) and visualized them in Figure 3. The figure illustrates how the regularization parameter $\lambda$ effectively governs the degree of dispersion in the representations. Specifically, when using a small $\lambda$ value (e.g., 0.0), Sparse CL tends to produce relatively compact representations, as depicted in Figure 3(a). Conversely, with a larger $\lambda$ value (e.g., 0.1), Sparse CL tends to generate more diverse representations, as shown in Figure 3(d).

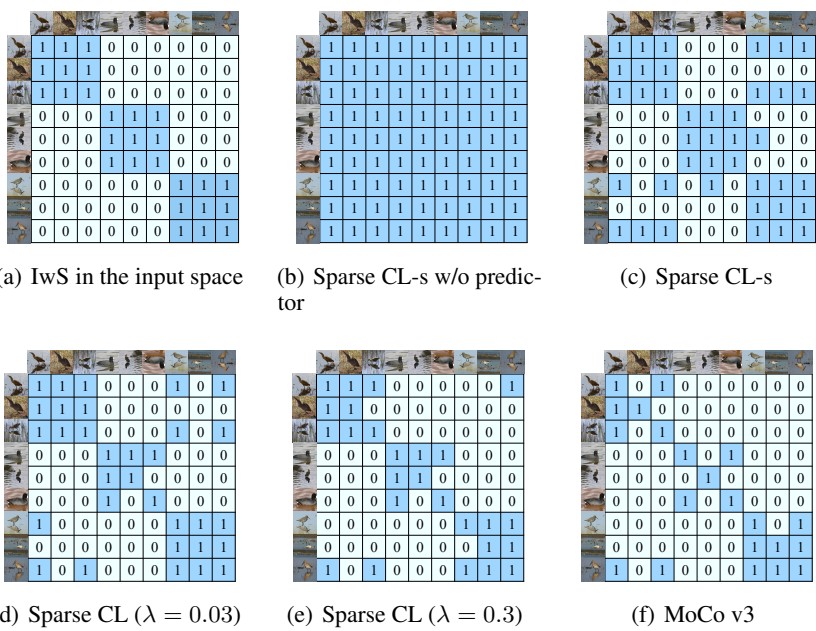

Figure 4: IwS visualization on the ImageNet-100 dataset. 1/0 indicates the same/different semantic information for the specified pair.

In addition to the PCA visualization, we also visualize the learned IwS in Figure 4. We pick 9 images from three challenging-to-distinguish classes (bittern, water hen, and red-backed sandpiper), with each class containing three images. The IwS in the input space is presented in Figure 4(a). We observe that when the predictor is removed from the network architecture in Sparse CL-s, it leads to an all-one IwS matrix, as shown in Figure 4(b), indicating a severe "collapse problem" (Jing et al., 2022). Surprisingly, upon reintroducing the predictor, the sparse structure re-emerges in Sparse CL-s, as visualized in Figure4(c). However, this level of sparsity remains insufficient compared to the sparsity of the IwS matrix in the input space. To analyze the impact of the regularization parameter $\lambda$, we gradually increased it from 0 to 0.3. This led to a significant increase in the sparsity of the IwS matrix, as depicted in Figures 4(d) and 4(e). Comparing with MoCo v3, which assumes the IwS matrix to be an identity matrix, as in Figure 4(f), our proposed Sparse CL outperforms MoCo v3 by producing learned IwS matrices in the representation space that more closely resemble those in the input space.

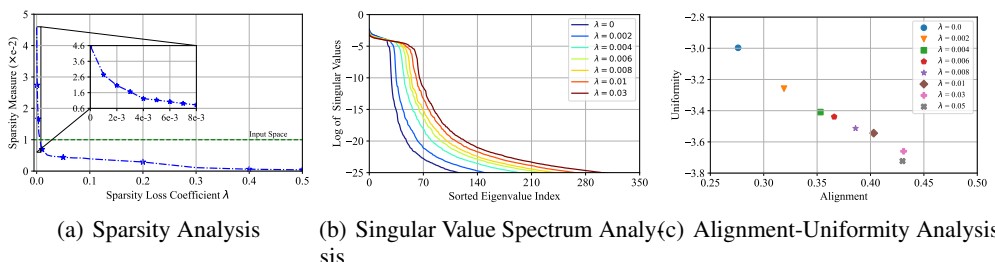

(a) Sparsity Analysis  (b) Singular Value Spectrum Analysis  (c) Alignment-Uniformity Analysis

Figure 5: Representations analysis w.r.t various sparse loss coefficients ($\lambda$) on the CIFAR-100 dataset.

**Quantitative Analysis on Learned Representations**   To gain a better understanding of how the proposed sparsity loss impacts representation learning, we conduct quantitative analyses on learned representations with respect to different choices of the regularization parameter $\lambda$ from three different perspectives. These analyses are performed using the CIFAR-100 dataset, and we extract representations from the predictor in the online encoder.

First, we conducted a sparsity analysis on the extracted representations using the proportion of value-one elements in the learned IwS matrix as the sparsity measure. The results are shown in Figure 5(a), where it is evident that a larger $\lambda$ leads to a smaller proportion of value-one elements, indicating increased sparsity in the learned representations.

Next, we perform a dimensional collapse analysis using the singular value spectrum (Jing et al., 2022) of the extracted representations. This analysis reveals that a larger $\lambda$ reduces the degree of dimensional collapse, as visualized in Figure 5(b).

Finally, we assess the alignment and uniformity of the representations using $\mathcal{L}$align and $-\mathcal{L}$uniform metrics, respectively, as proposed by Wang & Isola (2020). Figure5(c) illustrates that a larger $\lambda$ helps increase uniformity and slightly hamper alignment. This suggests that the sparsity penalty increases the uniformity of learned representations at the cost of a slight reduction in alignment.

**Sensitivity Analysis**   Figure 6 presents the top-1 accuracies of our proposed method on the CIFAR-100 dataset with varying hyperparameters, including the regularization parameter $\lambda$, the threshold $t_0$, and the temperature $\tau$. We explore the range of $\lambda$ from 0.0 to 0.5, and it is evident from Figure 6(a) that neither extremely small nor excessively large values of $\lambda$ result in optimal performance. This observation supports the advantage of our proposed Sparse CL over BYOL (Grill et al., 2020) and MoCo v3 (Chen et al., 2021), as these methods can be seen as extreme examples that use either a very small $\lambda$ (e.g., $\lambda = 0.0$) or an overly large $\lambda$, leading to under-sparse or over-sparse IwS matrices. In contrast, by carefully selecting an appropriate value of $\lambda$, our proposed Sparse CL can learn an appropriately sparse IwS in the representation space, thereby achieving better representations.

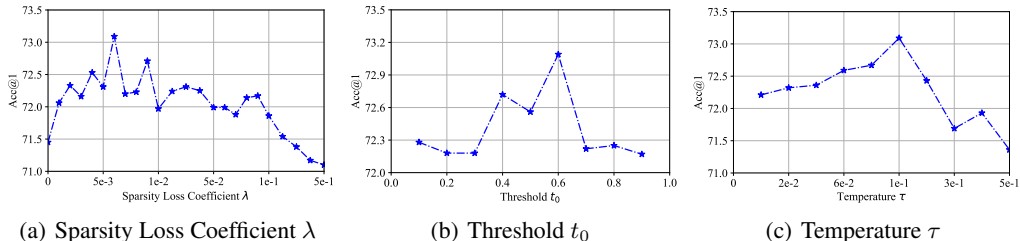

(a) Sparsity Loss Coefficient $\lambda$       (b) Threshold $t_0$       (c) Temperature $\tau$

Figure 6: Sensitivity analysis of the influence of different hyper-parameters on our proposed method.

The above analysis on regularization parameters $\lambda$ can be also applied to $t_0$ and $\tau$, with results collected in Figures 6(b) and 6(c), respectively. Similarly, too small/large $t_0$ would lead to an under-sparse/over-sparse IwS matrix in the representation space. For the temperature $\tau$, a too small $\tau$ (e.g., 0.01) would lead to vanishing gradients in the sigmoid activation in the Equation 3, while a too large $\tau$ (e.g., 0.5) can not enforce the $\mathbf{W}^{ab}$ and $\hat{\mathbf{W}}^{ab}$ to be binary matrices.

## 4   CONCLUSION

This paper unifies some self-supervised learning methods and offers a better understanding of them from the perspective of instance-wise similarity. Using this perspective, we identify the key issues of Siamese networks and constrastive learning. Specifically, contrastive learning assumes the IwS matrix to be an identity or extremely sparse matrix, while the Siamese network does not utilize any sparse structure and may produce collapsing solutions. To overcome these issues, we propose Sparse CL, a new self-supervised approach that could learn an appropriately sparse IwS matrix in the representation space and thus better representations. Extensive numerical studies on ImageNet and CIFAR datasets empirically demonstrate the effectiveness of our proposed method. One limitation of our work is that the proposed IwS is an instance-wise similarity measure, which hinders direct extensions of the proposed sparsity loss to more cases, such as the dimension-contrastive methods.

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

## A    EXPERIMENTAL SETTING

To achieve a fair comparison, we follow Zero-CL (Zhang et al., 2022), and employ ResNet-18 (He et al., 2016) as the encoder network for CIFAR-10/100 and ImageNet-100 datasets, and ResNet-50 (He et al., 2016) for the ImageNet-1k dataset. Note that on CIFAR-10/100 datasets, we employ commonly used tricks for low-resolution datasets by removing the first maxpool layer and modifying the first convolutional layer with kernel size 3 and strides 1 in ResNet-18 (Chen et al., 2020a; Zhang et al., 2022). In terms of image augmentation, we follow the pipeline of Barlow Twins (Zbontar et al., 2021) and Zero-CL (Zhang et al., 2022), which consist of random cropping, resizing to 224 × 224 (32 × 32 for CIFAR), horizontal flipping, color jittering, converting to gray-scale, Gaussian blurring, and solarization. During the pre-training phase, we use LARS optimizer (You et al., 2017) with a base learning rate $lr_1$, along with a cosine decay learning rate schedule (Loshchilov & Hutter, 2017) for all experiments. Specifically, our proposed Sparse CL are pre-trained 1000 epochs on CIFAR-10/1000 datasets, 400 epochs on the ImageNet-100 dataset, and 100/200 epochs on the ImageNet-1k dataset, respectively. We evaluate all experiments under a linear evaluation protocol by adding a linear classifier on top of fixed representations of ResNets pre-trained by Sparse CL. Specifically, the linear classifier is trained for 100 epochs by the SGD optimizer with a learning rate of $lr_2$ and a cosine learning rate scheduler. More detailed experimental settings are shown in Table 6.

Table 6: Experimental settings for various datasets in experiments.

| Datasets | CIFAR-10/100 | ImageNet-100 | ImageNet-1k |
|---|---|---|---|
| Epochs | 1000 | 400 | 100/200 |
| Batch Size | 256 | 1024 | 2048 |
| $lr_1$ | 0.2 | 0.1 | 0.05 |
| $lr_2$ | 0.3 | 0.1 | 0.15 |
| $t_0$ | 0.5/0.6 | 0.6 | 0.7 |
| $\tau$ | 0.1 | 0.1 | 0.1 |
| $\lambda$ | 1e-3/6e-3 | 0.03 | 0.04 |
| Feature Dimension | 2048 | 2048 | 4096 |
| Encoder Network | ResNet-18 | ResNet-18 | ResNet-50 |
| GPU Resources | 1 1080Ti GPU | 8 1080Ti GPUs | 8 A100 GPUs |

## B    TRAINING DYNAMICS ON THE CIFAR-100 DATASET

To empirically understand how Sparse CL addresses the trade-off between alignment loss and sparsity loss, we investigate their training dynamics. We plot the curve of alignment loss $\mathcal{L}_A$ and sparsity loss $\mathcal{L}_{sp}$ in logarithmic scale on the CIFAR-100 training dataset, as shown in Figure 7(a) and 7(b), respectively. We find the alignment loss decreases very fast at the early stage, and then converges smoothly, while the sparsity loss always keeps a relatively smooth decrease till the convergence. Besides, by enlarging the sparsity loss coefficient $\lambda$, although it slightly harms the convergence of alignment loss, it results in a significant decline in the curve of sparsity loss.

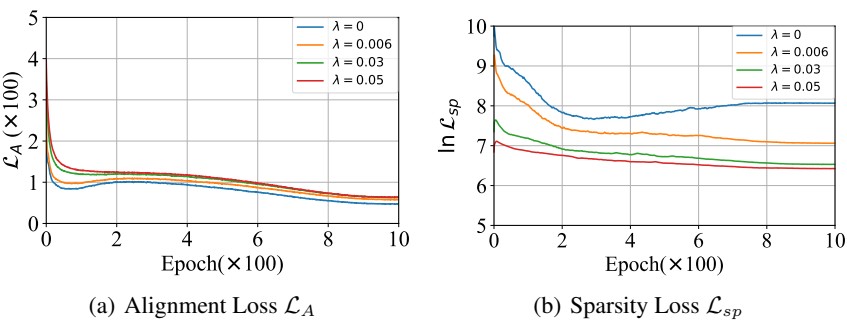

(a) Alignment Loss $\mathcal{L}_A$          (b) Sparsity Loss $\mathcal{L}_{sp}$

Figure 7: Training dynamics on the CIFAR-100 dataset.

## C ANALYSIS ON ARCHITECTURE COMPONENTS

Analyzing the role of asymmetric architecture components such as an extra predictor ('Pred'), momentum update ('Momem'), and stop gradient operator ('SG') in self-supervised learning is an important research topic in the machine learning community (Tian et al., 2021). To study the effect of these components in our proposed Sparse CL, we remove them step by step within Sparse CL architecture. As shown in Table 7, we can observe that the predictor and momentum update have a significant impact on improving linear evaluation performance, particularly in CIFAR-100 and ImageNet-100 datasets, while using the stop gradient operator only brings minor improvement.

Table 7: Ablation study of asymmetric architecture components, including applying the stop gradient operator ('SG'), the Momentum update ('Momem'), and an extra predictor ('Pred') on one of the two branches in our proposed Sparse CL.

| Methods | SG | Momem | Pred | CIFAR-10 | | CIFAR-100 | | ImageNet-100 | |
|---|---|---|---|---|---|---|---|---|---|
| | | | | Acc@1↑ | Acc@5↑ | Acc@1↑ | Acc@5↑ | Acc@1↑ | Acc@5↑ |
| Sparse CL | ✓ | ✓ | ✓ | **93.45** | **99.89** | **73.09** | **93.72** | **80.98** | **95.72** |
| Sparse CL | ✓ | ✓ | ✗ | 92.61 | 99.86 | 72.18 | 92.96 | 79.92 | 95.22 |
| Sparse CL | ✓ | ✗ | ✗ | 92.43 | **99.89** | 70.94 | 92.30 | 78.80 | 94.58 |
| Sparse CL | ✗ | ✗ | ✗ | 92.39 | 99.85 | 70.93 | 92.48 | 78.76 | 94.84 |

## D COMPARISON BETWEEN $\ell_1$ NORM AND $\ell_2$ NORM

In Sec. 2.3, to achieve binarization of IwS in the representation space, we apply $\ell_1$ norm to $\hat{\mathbf{W}}^{ab}$ and $\hat{\mathbf{W}}^{ba}$ in Equation 4. In this section, we replace the $\ell_1$ norm with $\ell_2$ norm, and experimental results are compared in Table 8. Sparse CL ($\ell_1$) performs better than Sparse CL ($\ell_2$) on CIFAR-10/100 datasets, and we explain that $\ell_1$ norm attributes to learning the sparse structure of the IwS matrix in the representation space, while $\ell_2$ norm cannot.

Table 8: Linear evaluation performance on CIFAR-10/100 datasets.

| Methods | CIFAR-10 | | CIFAR-100 | |
|---|---|---|---|---|
| | Acc@1↑ | Acc@5↑ | Acc@1↑ | Acc@5↑ |
| Sparse CL-s | 93.12 | 99.86 | 71.45 | 92.92 |
| Sparse CL ($\ell_1$) | **93.45** | **99.89** | **73.09** | **93.72** |
| Sparse CL ($\ell_2$) | 93.04 | 99.81 | 72.18 | 92.99 |

## E SENSITIVITY ON BATCH SIZE

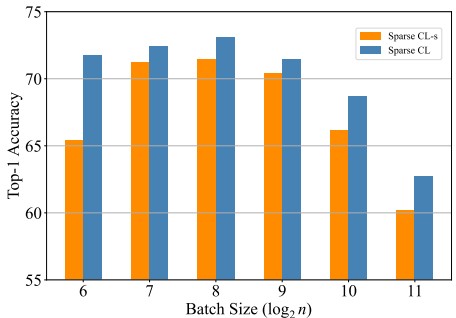

To investigate the impact of batch size when pre-training Sparse CL-s and Sparse CL from scratch, we vary the batch size from $2^6$ to $2^{11}$ and conduct experiments on the CIFAR-100 dataset. As visualized in Figure 8, we can observe that both Sparse CL-s and Sparse CL are sensitive to the batch size. Therefore, it is not recommended to use a too-large or too-small batch size on the relatively small datasets during the pre-training stage.

Figure 8: Linear evaluation of our proposed model trained with different batch size on the CIFAR-100 dataset.

