# OpenReview forum: "Rethinking Self-Supervise Learning: An Instance-wise Similarity Perspective"
_ICLR.cc/2024/Conference — ICLR 2024 Conference Withdrawn Submission_

### Official Review · Reviewer_dXtK · 2023-10-31

**Soundness:** 2 fair
**Presentation:** 3 good
**Contribution:** 2 fair
**Rating:** 5
**Confidence:** 4

**Summary:**

This paper proposes Sparse-CL, a new contrastive self-supervised learning method that, as opposed to most methods, allows positive pairs across views from different samples. The number of such positive pairs is regularized with a constraint on the instance-wise similarity matrix. Sparse-CL is evaluated on various linear classification benchmarks such as CIFAR and ImageNet, and demonstrates competitive performance in similar setups compared to concurrent methods.

**Strengths:**

1) The idea of considering different samples as positive pairs in a contrastive loss is interesting and fixes the issue of  repelling examples from similar concepts. There is an underlying graph of connections between concepts and the contrastive loss does not take that into account, which is a good motivation for this work.

2) The sparsity constraint is also a good idea. Indeed, discovering the graph might be very difficult and letting the system discover it with properly designed loss constraints seems to be a solution.

3) The results on small datasets are promising and the method achieves a very good performance against competitors.

**Weaknesses:**

1) I disagree that the IwS matrix of SimSiam isa matrix full of ones. In practice, the “critical issues” mentioned with SimSiam are not observed and I am not sure that it can be considered as a problem.

2) The results on ImageNet are good on a comparable setting, but far from being impressive. For example, Swav is compared without multi-crop, which is part of the method. Moreover, recent breakthroughs with the transformer architecture lead to much better results than what is reported in the paper. DINO reported 75% linear evaluation accuracy in 2021 and DINOv2 best model is at 86% accuracy.

3) Sparse-CL with lambda=0.0 performs 71.5% on Cifar-100, which is already better than every other method. How do you explain that ? Is the setup really comparable with other methods ?

4) Explanation in paragraphs Input space and Representation space are redundant and should be independent of the choice of method, here MoCo style method. Maybe just say: $\hat{W}$ is $W$ but in representation.

**Questions:**

Do you have a way of measuring if your method brings in practice in terms of distance between concepts in representation space, compared to classical methods ? Maybe using k-nn ? It might be possible that other methods already compute a graph of concepts automatically.

Would it be possible to adapt the method to work with redundancy reduction methods (Barlow Twins, VICReg) ?

---

### Official Review · Reviewer_TW8A · 2023-11-02

**Soundness:** 2 fair
**Presentation:** 2 fair
**Contribution:** 2 fair
**Rating:** 3
**Confidence:** 4

**Summary:**

This paper studies the effect of the regularization term on the IwS metric. By controlling the coefficient of the regularization term, the proposed Sparse CL method is capable of controlling the sparsity of the representation IwS. This paper shows that the method is effective on the downstream classification task.

**Strengths:**

1. This paper designs a loss function to control the sparsity of the IwS metric, which makes good use of inter-image information.
2. The empirical results and analysis prove the effectiveness of their method on classification tasks.

**Weaknesses:**

1. The IwS of the Siamese network seems to be wrong. Siamese network does not apply constraints on the non-diagonal items so it should not be an all-one matrix. It is shown that non-contrastive SSL implicitly reduces the similarity of off-diagonal samples [1].
2. The goal of SSL is to learn generalizable representation rather than improve classification performance. Therefore, the soundness of this paper could be further improved by providing experimental results on other downstream tasks like kNN, semantic segmentation, and object detection.
3. The authors provide neither methodology nor empirical comparison with existing inter-image self-supervised learning methods like [2].

Ref:

[1] Zhuo, Zhijian, et al. "Towards a Unified Theoretical Understanding of Non-contrastive Learning via Rank Differential Mechanism." ICLR, 2023.

[2] Xie, Jiahao, et al. "Delving into inter-image invariance for unsupervised visual representations." IJCV, 2022.

**Questions:**

From the sensitivity analysis, we can see that the performance is sensitive to the parameter. Given input images, is it possible to estimate the input IwS so that we know the desired sparsity of the representation IwS?

---

### Official Review · Reviewer_LXXA · 2023-11-02

**Soundness:** 2 fair
**Presentation:** 2 fair
**Contribution:** 2 fair
**Rating:** 5
**Confidence:** 5

**Summary:**

The paper proposes to understand self-supervised learning from the perspective of instance-wise similarity (IwS). From this perspective, the paper identifies the limitations in current self-supervised learning approaches, including contrastive learning and Siamese methods. To address the limitations, the paper introduces sparse contrastive learning, that learns an appropriately sparse IwS matrix in the representation space. The proposed method is validated through experiments on ImageNet and CIFAR datasets, showing superior performance compared to other state-of-the-art methods.

**Strengths:**

1. The work is well-motivated, aiming to bridge the discrepancy between IwS matrices in input and representation spaces.
2. The paper is well-organized, and the proposed method is explained with visual illustrations which aid in understanding the concept of IwS and the proposed Sparse CL approach.
3. The authors provide extensive experiments on standard classification benchmarks, including CIFAR-10, CIFAR-100, ImageNet-100, and ImageNet-1k, which substantiate the claimed benefits of Sparse CL.

**Weaknesses:**

1. Contrastive methods in self-supervised learning adopt instance discrimination as the pretext task, and the focus of this line of research (and most previous methods) is how to handle the positive and negative pairs, or the diagonal and off-diagonal entries in the IwS matrix, respectively. However, the authors claim that studying from the perspective of IwS provides a novel framework, which might not be true. In addition, the proposed method fails to deal with false positives, as there might be 0s in the diagonal of the IwS matrix due to semantic inconsistency caused by strong data augmentation. This scenario should be taken into consideration since this paper focuses on IwS.
2. A theoretical analysis can be conducted to analyze the alignment and sparsity terms of Sparse CL loss. The InfoNCE loss, used in SimCLR, MoCo and other contrastive methods, can also be decomposed into two terms similar to the proposed loss. The authors should theoretically discuss the relationships between these losses to better demonstrate the advantages of the proposed method.
3. The paper could benefit from a broader evaluation on other tasks beyond classification to further validate the generalization ability of the proposed method. For example, the pretrained model can be transferred to object detection and segmentation tasks, which is commonly used to evaluate the performance of self-supervised learning methods.
4. Minors: Fig.1(d) shows an all-one IwS matrix for Siamese methods, which is not appropriate, as this is only the situation of mode collapse and the Siamese methods have already addressed this problem. Fig.4 shows a binary 0/1 matrix of Sparse CL in representation space, while Eqn.3 computes a continuous similarity value between 0 and 1. How is the above conversion made?

**Questions:**

Please check the weakness.